# From roads to biobanks: Roadkill animals as a valuable source of genetic data

**Manuel Alejandro Coba-Males**[1☯], **Pablo Medrano-Vizcaíno**[2,3,4☯], **Sandra Enríquez**[1], **David Brito-Zapata**[4,5], **Sarah Martin-Solano**[6], **Sofía Ocaña-Mayorga**[7], **Gabriel Alberto Carrillo-Bilbao**[1], **Wilmer Narváez**[1], **Jaime Antonio Salas**[8], **Jazzmín Arrivillaga-Henríquez**[1], **Manuela González-Suárez**[2]*, **Ana Poveda**[1]*

**1** Grupo de Investigación en Biodiversidad, Zoonosis y Salud Pública (GIBCIZ), Instituto de Investigación en Zoonosis (CIZ), Facultad de Ciencias Químicas (FCQ), Universidad Central del Ecuador, Quito, Ecuador, **2** Ecology and Evolutionary Biology, School of Biological Sciences, University of Reading, Reading, United Kingdom, **3** Universidad Regional Amazónica IKIAM, Grupo de Investigación Población y Ambiente, Tena, Ecuador, **4** Red Ecuatoriana para el Monitoreo de Fauna Atropellada-REMFA, Quito, Ecuador, **5** Instituto iBIOTROP, Museo de Zoología & Laboratorio de Zoología Terrestre, Universidad San Francisco de Quito USFQ, Quito, Ecuador, **6** Grupo de Investigación en Sanidad Animal y Humana (GISAH), Carrera Ingeniería en Biotecnología, Departamento de Ciencias de la Vida y la Agricultura, Universidad de las Fuerzas Armadas —ESPE, Sangolquí, Ecuador, **7** Centro de Investigación para la Salud en América Latina, Facultad de Ciencias Exactas y Naturales, Pontificia Universidad Católica del Ecuador, Quito, Ecuador, **8** Facultad de Ciencias Naturales, Carrera de Biología, Universidad de Guayaquil, Guayaquil, Ecuador

☯ These authors contributed equally to this work.
* manuela.gonzalez@reading.ac.uk (MGS); apoveda@uce.edu.ecapo (AP)

**Data Availability Statement:** All relevant data are within the manuscript and its Supporting Information files.

## Abstract

To protect biodiversity we must understand its structure and composition including the bacteria and microparasites associated with wildlife, which may pose risks to human health. However, acquiring this knowledge often presents challenges, particularly in areas of high biodiversity where there are many undescribed and poorly studied species and funding resources can be limited. A solution to fill this knowledge gap is sampling roadkill (animals that die on roads as a result of collisions with circulating vehicles). These specimens can help characterize local wildlife and their associated parasites with fewer ethical and logistical challenges compared to traditional specimen collection. Here we test this approach by analyzing 817 tissue samples obtained from 590 roadkill vertebrate specimens (Amphibia, Reptilia, Aves and Mammalia) collected in roads within the Tropical Andes of Ecuador. First, we tested if the quantity and quality of recovered DNA varied across roadkill specimens collected at different times since death, exploring if decomposition affected the potential to identify vertebrate species and associated microorganisms. Second, we compared DNA stability across taxa and tissues to identify potential limitations and offer recommendations for future work. Finally, we illustrate how these samples can aid in taxonomic identification and parasite detection. Our study shows that sampling roadkill can help study biodiversity. DNA was recovered and amplified (allowing species identification and parasite detection) from roadkill even 120 hours after death, although risk of degradation increased overtime. DNA was extracted from all vertebrate classes but in smaller quantities and with lower quality from amphibians. We recommend sampling liver if possible as it produced the highest amounts of DNA (muscle produced the lowest). Additional testing of this approach in areas with different environmental and traffic conditions is needed, but our results show that

**Funding:** The authors would like to thank to: Corporación Ecuatoriana para el Desarrollo de la Investigación y Academia - CEDIA for the financial support given to the present research, development, and innovation work through its CEPRA program, especially for the Estudio de parásitos y microbioma de fauna silvestre en dos de las zonas más biodiversas del planeta: Los Andes Tropicales y Chocó-Darién en Ecuador fund (https://cedia.edu.ec/servicio/fondo-idi-universidades/cepra-xvi-2022/); Dirección de Investigación de la Universidad Central del Ecuador for Proyecto Senior 2021 fund (https://www.uce.edu.ec/web/di), and Reading University for SBS Seed Fund and a University International PhD studentship to PMV (https://www.reading.ac.uk/. The funders had no role in study design, data collection and analysis, decision to publish, or preparation of the manuscript.

**Competing interests:** The authors have declared that no competing interests exist.

sampling roadkill specimens can help detect and potentially monitor biodiversity and could be a valuable approach to create biobanks and preserve genetic data.

## Introduction

Characterizing biodiversity including bacteria and microparasites is key to ensure planetary and human health. However, acquiring this knowledge often presents challenges including ethical (the need to capture, handle, and take samples from living organisms without causing damage or death), logistical (the difficulty and cost of safely capturing and handling wildlife), and methodological (the challenges for identifying and quantifying biodiversity in less studied regions and from microscopic bacteria and parasites). In biodiverse and remote areas where funding and research infrastructure are more limited, these challenges can be particularly significant. This knowledge gap could be partly reduced by capitalizing on a potential source of information: the animals that die on roads after collision with circulating vehicles, commonly referred to as roadkill.

Human infrastructures, including roads, have important impacts on wildlife by hindering movement (for food, mates, and refuge) and also causing direct mortality [1]. These aspects can lead to population isolation, loss of genetic diversity, and in some cases even increased extinction risk [2, 3]. However, animals that die on roads can be valuable resources for research. Roadkill specimens can capture the (primarily vertebrate) biodiversity of a particular location and give information about species distributions, behavior, diet, and other ecological and physiological characteristics [4]. Unfortunately, in some cases the required taxonomic identification can be difficult via direct examination due to the state of the specimen, these can decompose or present damage caused by the collision and subsequent vehicles driving over. As an alternative, collecting tissue from specimens could allow their identification via DNA sequencing [5].

Tissue collected from roadkill specimens can be also provide very valuable information about associated microorganisms such as fungi, protozoan, bacteria, and viruses [6–10]. This information is important because these microorganisms can be pathogenic for wildlife and thus, relevant for wildlife conservation and management. In addition, wildlife can act as reservoirs of human pathogens posing a health problem [9]. In this case, roadkill sampling used to detect pathogens could align well with the One Health framework that integrates animal, human, and environmental health [11]. Pathogen data can be obtained using traditional sampling methods (from live animals), but these are often invasive and can negatively affect wildlife, so there is a need for improved methods [12]. Alternative methods based on non-invasive sampling from excretions or feces avoid impact but often the amount of extracted DNA is low [13, 14]. Sampling roadkill could be a useful approach in areas where roads exist, offering no sampling impact and potentially higher DNA yield. However, first we need to understand how the source of the sample (which taxa and tissue) and the degree of decomposition of the roadkill specimen affect DNA quantity and quality (purity and integrity) [15, 16].

Here, we address these questions by analyzing 817 samples from 590 roadkill specimens representing amphibians, reptiles, birds, and mammals found dead on roads of the Napo province in Ecuador. Whenever possible we collected samples from multiple tissues of the same individual and report the obtained DNA quantity and quality. In addition, we explore the value of collected samples for molecular identification of specimens and for detection of microorganisms, in particular of the protozoan *Leishmania* spp. These parasites can affect

humans, domestic animals like dogs and cats [17], wild mammals [18, 19], and likely birds and reptiles [19–21].

## Materials and methods

### Study area and roadkill sampling

This work was conducted in the Amazonian province of Napo in Ecuador. We surveyed roads surrounded by four protected areas: Antisana Ecological Reserve, Sumaco-Napo-Galeras National Park, Cayambe-Coca National Park, and Colonso Chalupas Biological Reserve. This area is located across an altitudinal gradient from 300 to 3000 m.a.s.l. and a climatic gradient, with annual mean temperature varying from 4.63 to 23.7˚C and annual precipitation from 1100 to 3400 mm. The study area is described in more detail in previous publications [22, 23].

During 100 non-consecutive days from the 19th of September 2020 to the 23rd of March 2021, we monitored 240 km of primary and secondary roads from a car circulating at an average speed of 40 km/h. Each time we found a carcass, we stopped the car, collected the whole individual or muscle samples if the state of the carcass preventing subsequent dissection for tissue extraction. All collected material was stored in a container with dry ice until the end of the day when all was taken to a laboratory. In the laboratory whole individuals were dissected aiming to extract samples of muscle, brain, liver, heart, intestine, blood, blood vessel, bone, lung, skin, and spleen. The condition of some specimens did not allow collecting samples from all tissues. All samples were then frozen at -80˚C until DNA extraction (see below).

For each specimen we estimated the time since death, *TimeDeath*, in time blocks representing: 0, 12, 24, 36, 48, 60, 72, 96, or 120 hours. Observers had been previously trained to estimate time since death by marking roadkill specimens which were revisited daily for ten days to characterize degradation [22, 24].

All samples were collected with permission from the Ministerio del Ambiente, Agua y Transición Ecológica from Ecuador (MAATE): Estudio e identificación molecular de parásitos y microbioma presentes en fauna silvestre del Ecuador, No. MAAE-DBI-CM-2021-0215, MAAE-ARSFC-2021-1862 and MAAE-ARSFC-2020-0791.

### DNA extraction and evaluation of quantity and quality

We extracted DNA from tissues using a commercial kit (PureLink™ Genomic DNA Mini Kit, from Invitrogen by Thermo Fisher Scientific) following the manufacturer's instructions. We measured the quantity of nucleic acids using a Nanodrop 2000 (Thermo Fisher Scientific) with values transformed into ng/µL (full dataset available as S1 File). To assess DNA quality, we considered two parameters: the purity given by the ratio $A_{260}/A_{280}$, and integrity. Integrity was only evaluated in intestine tissue samples as we were particularly interested in microbiome and gastrointestinal parasites. We qualitatively classified integrity based on the size of the smear observed in gel electrophoresis [25] as: "minor degradation" (sizes over 6000 bp), "medium degradation" (sizes from 1500 bp to 6000 bp), or "high degradation" (sizes below 1500 bp or not visualized DNA).

**Statistical analyses.** All statistical analyses were conducted in R version 4.2.1 [26]. We tested if DNA quantity and purity were influenced by estimated *TimeDeath* and the taxonomic class of the roadkill specimen using linear regression models with the function lm in base R. We fitted models with only additive terms as well as models with an interaction term. We report the most parsimonious model (i.e., if the interaction term was not significant, we report results from the additive model). To avoid pseudoreplication caused by including data from multiple tissues obtained from the same specimen, we selected the sub-sample with higher DNA concentration (ng/µL). Results were qualitatively the same if we instead used the mean

value calculated across tissues from the same specimen (S1 Table). In addition, to test the robustness of results to the potential non-independence of data obtained from related organisms with shared evolutionary history, we also fitted linear mixed effects regression models using the function lmer from the lme4 package version 1.1–27.1 [27] with taxonomic order and family as nested values random effects. Because not all specimens could be identified to order or family levels, mixed effect models included fewer specimens.

To test if DNA quantity and purity were influenced by the sampled tissue, we focused on tissues sampled from at least 50 specimens (obtaining samples from some tissues was difficult due to their size and degradation). In this case we analyzed all samples, including replicates from the same specimen, and thus, to avoid pseudoreplication we fitted mixed effects regression models with specimen ID as a random factor using the function lmer from the lme4 package version 1.1–27.1.

To meet assumptions of normality and homoscedasticity of residuals we $\log_{10}$ transformed DNA quantity and purity values. Model assumptions (linearity, normality and homoscedasticity of residuals, and lack of outliers) were visually checked using the function check_model from the performance package version 0.7.2 [28]. This function also returns Variance Inflation Factors (VIF) that were used to test for collinearity among predictors. We used the function emtrends from the package emmeans version 1.6.1 [29] to estimate the estimated marginal means reported as model output.

## Molecular identification of vertebrates and parasites

For the identification of vertebrate specimens, we amplified 700 bp of mitochondrial *cytochrome C oxidase subunit 1* (*COI*) using 10 pmol/μL of two primer cocktails. The forward cocktail (C_VF1LFt1) contained VF1_t1:VF1d_t1:LepF1_t1:VF1i_t1 (1:1:1:3). The reverse cocktail (C_VR1LRt1) contained VR1_t1:VR1d_t1:LepRI_t1:VR1i_t1 (1:1:1:3) (Table 1). Analyses focused on eight specimens (seven reptiles and one amphibian) that could not be identified through direct observation (i.e., traditional taxonomic identification) as our goal was to generate a complete identified dataset.

The polymerase chain reaction (PCR) was performed with Platinum™ Taq DNA Polymerase from Invitrogen as described by Hebert et al. [32] with some modifications; 2.5 μL buffer 10 X, 1.15 μL (2.3 mM) $MgCl_2$, 0.12 μL (0.046 mM) dNTPs, 0.25 μL (0.1 mM) of each cocktail of oligonucleotides (C_VF1LFt1 and C_VR1LRt1), 7.5 μL (0.3 mg/mL) of BSA, 0.5 U of Taq polymerase, and 100 ng of DNA per reaction in a final volume of 25 μL. PCR amplification was performed with an initial denaturation step (94°C, 1 min), followed by 5 cycles of denaturation (94°C, 1 min), annealing (50°C, 40 sec), and polymerization (72°C, 1 min), followed of 30 cycles of denaturation (95°C, 15 sec), annealing (54°C, 20 sec) and polymerization (72°C, 45 sec), and a final step of 10 min at 72°C.

To detect *Leishmania* spp. we amplified a small subunit (18S) of ribosomal RNA (*18S SSU-rRNA*) from liver samples of roadkill specimens [33] using universal primers (Table 1). We first used the external primers SLF/S762R (primary amplification) and the internal primers S825F/SLIR targeting a ∼959 bp product of the first half (Nt) of the *18S SSU-rRNA* locus [31]. The polymerase chain reaction (PCR) was performed with Platinum™ Taq DNA Polymerase from Invitrogen with 2.5 μL buffer 10 X, 0.75 μL (1.5 mM) $MgCl_2$, 0.5 μL (0.2 mM) dNTPs, 1 μL (0.4 μM) of each oligonucleotide, 0.5 U of Taq polymerase, and 100 ng of DNA for reaction in a final volume of 25 μL. PCR amplification was as described previously for primary and secondary *18S SSU-rRNA* [31].

We visualized the PCR products by DNA electrophoresis on a 1.5% agarose gel. Positive amplicons, identified by their size (˜ 700 bp for the *COI* gene and 959 bp for the *18 SSU-rRNA*

**Table 1. PCR primers used in this study to identify vertebrate roadkill specimens and *Leishmania* spp.**

| | *COI* | |
|---|---|---|
| Name | Primer sequence 5'-3' | Reference |
| LepF1_t1 | TGTAAAACGACGGCCAGTATTCAACCAATCATAAAGATATTGG | [30] |
| VF1_t1 | TGTAAAACGACGGCCAGTTCTCAACCAACCACAAAGACATTGG | |
| VF1d_t1 | TGTAAAACGACGGCCAGT TCTCAACCAACCACAARGAYATYGG | |
| VF1i_t1 | TGTAAAACGACGGCCAGTTCTCAACCAACCAIAAIGAIATIGG | |
| LepR1_t1 | TGTAAAACGACGGCCAGTATTCAACCAATCATAAAGATATTGG | |
| VR1_t1 | TGTAAAACGACGGCCAGTTCTCAACCAACCACAAAGACATTGG | |
| VR1d_t1 | TGTAAAACGACGGCCAGT TCTCAACCAACCACAARGAYATYGG | |
| VR1i_t1 | TGTAAAACGACGGCCAGTTCTCAACCAACCAIAAIGAIATIGG | |
| | *18S SSU rRNA* | |
| Name | Primer sequence 5'-3' | Reference |
| SLF | GCTTGTTTCAAGGACTTAGC | [31] |
| S762R | GACTTTTGCTTCCTCTAATG | |
| S823F | CGAACAACTGCCCTATCAGC | |
| S662R | GACTACAATGGTCTCTAATC | |
| S825F | ACCGTTTCGGCTTTTGTTGG | |
| SLIR | ACATTGTAGTGCGCGTGTC | |

gene), were sequenced in two directions: forward and reverse (Sanger method) by Macrogen Inc. (South Korea). Resulting sequences were trimmed and edited with Bioedit and MEGA 11 software [34, 35] to perform pairwise alignment, assembling the contig, and creating consensus sequences which were then analyzed by BLAST using the blastn algorithm in NCBI (National Center for Biotechnology Information) [36]. To assign sequences to *Leishmania* spp. and relevant vertebrate genera we used a threshold of 85% for identity and of 90% for cover.

## Results

We obtained 817 samples representing 590 unique roadkill specimens (Table 2, all data in S1 File). From 516 specimens we collected samples from a single tissue; the other 74 specimens were represented by samples from 2–7 different tissues.

Nucleic acids were successfully extracted from 812 samples representing 586 unique roadkill specimens with time since death (*TimeDeath*) varying between 0 and 120 hours. No DNA

**Table 2. Summary of the vertebrate roadkill samples analyzed in this study.**

| | Muscle | Brain | Liver | Heart | Intestine | Blood | Blood vessel | Bone | Lung | Skin | Spleen | |
|---|---|---|---|---|---|---|---|---|---|---|---|---|
| **Mammalia** | 100 | 1 | 4 | 3 | 2 | 0 | 2 | 2 | 4 | 2 | 2 | 122 |
| **Aves** | 114 | 21 | 19 | 26 | 21 | 3 | 5 | 0 | 19 | 2 | 0 | 230 |
| **Amphibia** | 116 | 1 | 3 | 2 | 3 | 0 | 6 | 0 | 1 | 0 | 0 | 132 |
| **Reptilia** | 136 | 25 | 31 | 26 | 27 | 0 | 57 | 5 | 15 | 4 | 2 | 328 |
| **TOTAL** | 466 | 48 | 57 | 57 | 53 | 3 | 70 | 7 | 39 | 8 | 4 | **812** |

Number of samples collected from different tissues of roadkill specimens collected in the Napo province of Ecuador. Data are separated by the four taxonomic classes of vertebrates represented in this study. Detailed dataset available as S1 File.

was obtained after extraction for four individuals with single tissue samples. In addition, no DNA was obtained from the brain sample from one specimen, but extractions were successful for the five other tissues obtained from that individual. Table 2 summarizes the samples analyzed.

## Effect of time since death on DNA quantity and purity

We obtained DNA from specimens of all sampled taxonomic classes including animals estimated to have died 120 hours before collection. The four specimens from which no DNA was obtained were estimated to have died between 12 and 72 hours before collection. DNA quantity and purity were similar among birds, mammals and reptiles, but Amphibia specimens returned significantly lower amounts and less pure DNA (Fig 1 and Table 3).

DNA quantity was significantly and negatively associated with the estimated time since death in birds but not in other taxonomic classes (Fig 2 and Table 3). DNA purity was not affected by the estimated time since death (Table 3).

## DNA quantity and purity from different tissues

We collected 703 samples from five different tissues represented at least by 50 samples: muscle (n = 466), blood vessels (n = 70), heart (n = 57), liver (n = 57), and intestine (n = 53); (see Table 2). DNA quantity varied among the compared tissues, with liver returning significantly more and muscle generally less (Fig 3 and Table 4). There were no differences in DNA purity among tissues (Table 4).

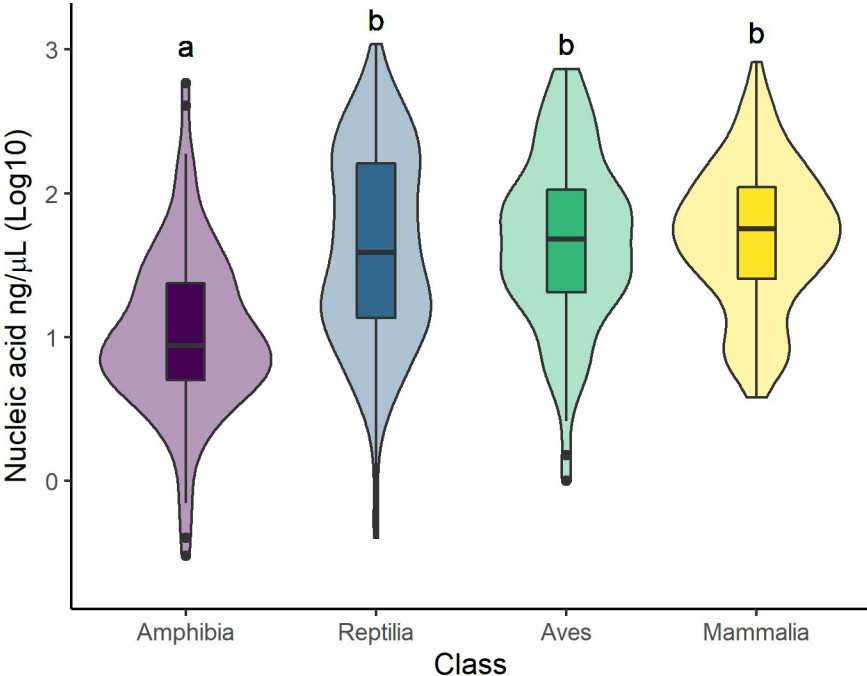

**Fig 1. DNA quantification by taxonomic class.** Variation in the amount of DNA extracted from 817 tissue samples obtained from specimens of four vertebrate classes collected during roadkill surveys in the Napo region of Ecuador from 2020 to 2021. Different letters above the plot indicate significant differences between the groups.

**Table 3. Linear regression models testing how DNA quantity and purity varied by class and time since death.**

| Predictor | Marginal means | SE | 95% CI |
|---|---|---|---|
| *Quantity of DNA (log10) (N = 586, $R^2$ = 0.18)* | | | |
| Amphibia | 1.010 | 0.0544 | 0.906–1.120 |
| Reptilia | 1.610 | 0.0408 | 1.531–1.690 |
| Aves | 1.640 | 0.0494 | 1.542–1.740 |
| Mammalia | 1.720 | 0.0574 | 1.604–1.830 |
| Amphibia *TimeDeath* | -0.002 | 0.0028 | -0.0072–0.0038 |
| Reptilia *TimeDeath* | -0.001 | 0.0015 | -0.0031–0.00263 |
| Aves *TimeDeath* | -0.009 | 0.0023 | -0.0136–-0.0047 |
| Mammalia *TimeDeath* | -0.004 | 0.0023 | -0.0081–0.00091 |
| *Purity of DNA (log10) (N = 586, $R^2$ = 0.02)* | | | |
| Amphibia | 0.217 | 0.0067 | 0.204–0.230 |
| Reptilia | 0.245 | 0.0050 | 0.235–0.255 |
| Aves | 0.243 | 0.0061 | 0.231–0.255 |
| Mammalia | 0.245 | 0.0070 | 0.231–0.259 |
| *TimeDeath* | -0.0002 | 0.0001 | -0.0005–0.00001 |

Linear regression results show how the amount and purity of DNA ($log_{10}$ scale) were influenced by the estimated time since death (*TimeDeath*) and the taxonomic class. Samples were collected during roadkill surveys in the Napo region of Ecuador from 2020 to 2021. We report estimated marginal means for each predictor and the interaction term (when interactions were significant), their standard error (SE), and their 95% confidence intervals (95% CI). For each model, we also report the number of specimens (*N*) for which data were available and the adjusted $R^2$ of the model. Results were qualitatively the same when accounting for evolutionary relationships in random effects models and when considering the mean of DNA amount per specimen (when multiple tissues were analyzed) instead of the best value (S1 Table).

## DNA integrity

DNA quantity ranged from 0.2 to 665 ng/μL among the 53 intestine samples analyzed, with a non-significant tendency for DNA to be more degraded in samples with longer times since death (Fig 4A), probably due to exposure to environmental conditions such as rain or high temperatures. Samples scored as high level of degradation yield significantly lower quantities of DNA (Fig 4B).

## Successful amplification of specific targets by PCR

We successfully amplified the targeted region of *cytochrome C oxidase subunit 1* (*COI*). in the eight tested samples from reptile and amphibian specimens (Fig 5B) despite the variability in the integrity of the extracted genomic DNA (Fig 5A and Table 5). All cover values were >95%, but identity values were lower potentially due to lack of existing sequences in for these species in the NCBI repository. Tentatively, we have used identity values ≥ 85% to assign the specimen to genera (Table 6).

Amplification was successful for *Leishmania* spp. in nine of the 57 liver samples (Fig 6 and Table 7). The match to the *Leishmania* genus had high certainty with cover ≥ 98% and identity ≥ 99%. Samples were best matched to the species *L. amazonensis*, in agreement with previous records from Ecuador [37–39].

While additional testing of the potential for using roadkill species for taxonomic identification and parasitology is needed, our results are promising. We were able to amplify gene

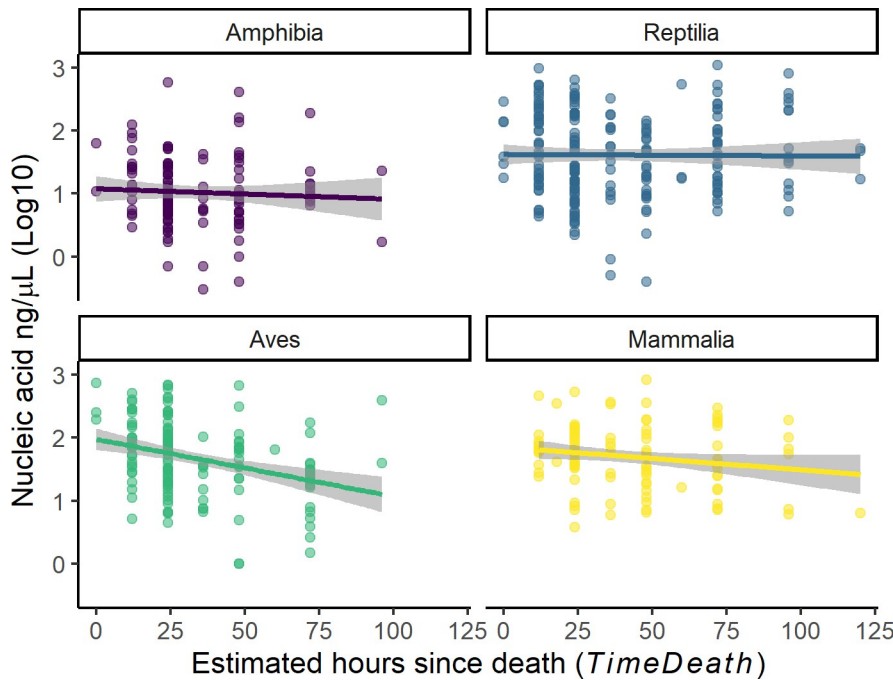

**Fig 2. Variation in the obtained quantity DNA from roadkill specimens at different estimated times since death across the four studied taxonomic classes.** Symbols represent the observed data with the line and shaded area representing the estimated relationship and 95% CI respectively from a linear regression model (Table 3). Samples were collected during a roadkill survey in the Napo region of Ecuador from 2020 to 2021.

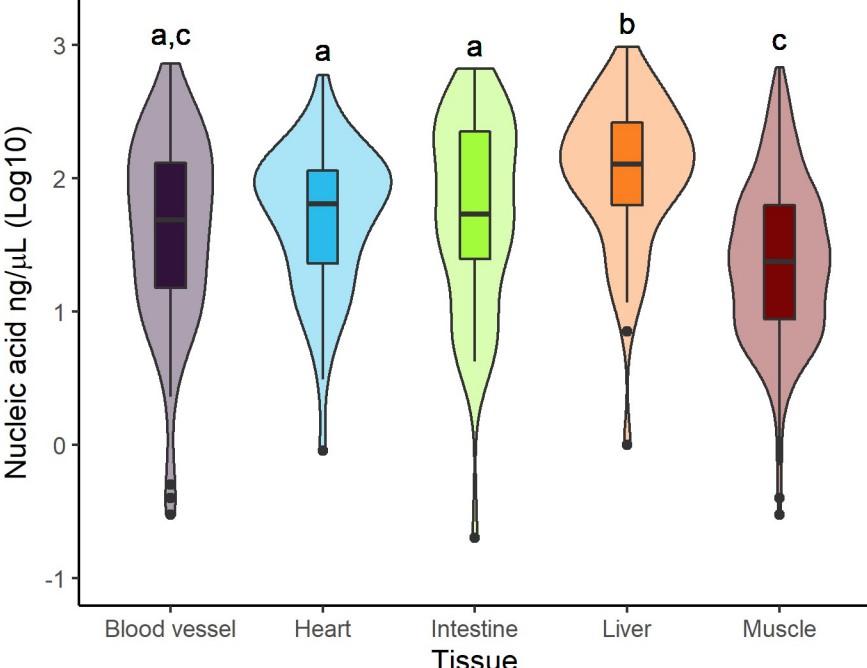

**Fig 3. DNA quantification by tissue.** Variation in the amount of DNA extracted from different tissues collected from roadkill vertebrates during surveys in the Napo region of Ecuador from 2020 to 2021. Different letters above the plot indicate significant differences between the groups.

**Table 4. Linear regression models testing how DNA quantity and purity varied by tissue.**

| Predictor | Marginal means | SE | 95% CI |
|---|---|---|---|
| *Quantity of DNA (log10) (N = 703, conditional R² = 0.51, marginal R² = 0.21)* | | | |
| **Amphibia** | 1.27 | 0.066 | 1.14–1.40 |
| **Reptilia** | 1.70 | 0.045 | 1.61–1.78 |
| **Aves** | 1.80 | 0.054 | 1.69–1.90 |
| **Mammalia** | 1.92 | 0.069 | 1.79–2.06 |
| **Blood vessel** | 1.58 | 0.070 | 1.44–1.72 |
| **Heart** | 1.66 | 0.075 | 1.51–1.81 |
| **Intestine** | 1.72 | 0.077 | 1.56–1.87 |
| **Liver** | 2.00* | 0.075 | 1.86–2.15 |
| **Muscle** | 1.39* | 0.027 | 1.34–1.44 |
| *Purity of DNA (log10) (N = 703, conditional R² = 0.10, marginal R² = 0.03)* | | | |
| **Amphibia** | 0.22 | 0.009 | 0.204–0.240 |
| **Reptilia** | 0.25 | 0.006 | 0.235–0.259 |
| **Aves** | 0.24 | 0.007 | 0.229–0.257 |
| **Mammalia** | 0.25 | 0.010 | 0.228–0.267 |
| **Blood vessel** | 0.24 | 0.011 | 0.222–0.265 |
| **Heart** | 0.27 | 0.012 | 0.242–0.288 |
| **Intestine** | 0.22 | 0.012 | 0.192–0.240 |
| **Liver** | 0.24 | 0.012 | 0.217–0.263 |
| **Muscle** | 0.24 | 0.004 | 0.227–0.243 |

Linear mixed effect regression results show how the amount and purity of DNA ($log_{10}$ scale) extracted from roadkilled specimens were influenced by the analyzed tissue and taxonomic class. Samples were collected during roadkill surveys in the Napo region of Ecuador completed in 2020–2021. We report estimated marginal means for each predictor and the interaction term (when interactions were significant), and their 95% confidence intervals (95% CI). For each model, we also report the number of specimens (*N*) for which data were available and the conditional and marginal $R^2$ of the model.

**Fig 4. DNA integrity.** Variation in the degradation of DNA extracted from intestine tissues collected at varying times time since death (A) and that yield different quantities of DNA (B). Samples were obtained from roadkill vertebrates during surveys in the Napo region of Ecuador completed in 2020–2021. Different letters above the plot indicate significant differences between the groups.

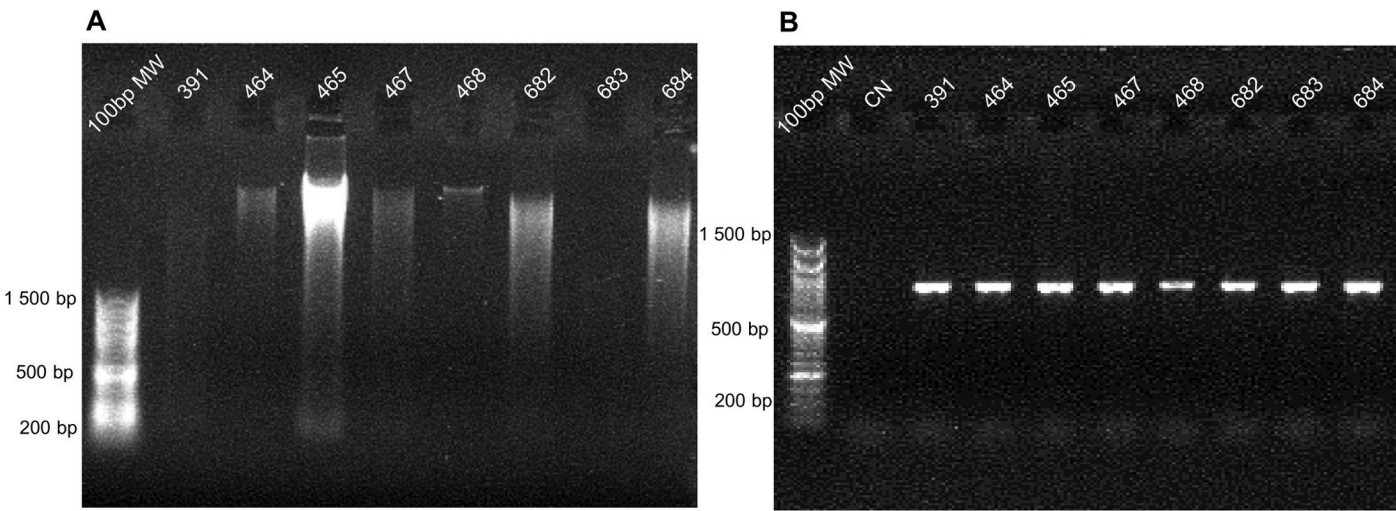

**Fig 5. Amplification of the targeted *COI* region for vertebrate taxonomic identification.** DNA electrophoresis from the eight tested vertebrate samples containing isolated genomic DNA (A) and the amplified *COI* (B). Molecular weight (MW); Negative control (CN). The identification number of each sample (ID) is shown in the top line.

fragments from DNA samples with concentration as low as 7.1 ng/μL (sample 209, Fig 6 and Table 8), and with high degradation (samples 391 and 683, Fig 5; samples 209 and 819, Fig 6).

## Discussion

Our results show that tissues samples collected from roadkill wildlife can offer a valuable source of genetic data and thus, contribute to our understanding of biodiversity. Samples can help identify vertebrate species and be use for detection, and potentially monitoring, of micro-organisms including those that can pose risks to human health. While the necessary genetic analyses can be costly (even if rapidly becoming more affordable), sampling roadkill is a simple, relatively low cost, low risk, and low impact (not requiring capturing of wildlife) method to gather tissue from wildlife. While not all animals cross roads or become roadkill, we know that many species are susceptible [40–42] and thus, this sampling approach can provide access to a large diversity of species. The resulting data could contribute to gain much needed understanding of existing biodiversity in less-studied areas and could be used in combination with other approaches such as environmental DNA (eDNA) and metabarcoding [43, 44].

In our study region, a tropical environment with relatively high humidity and non-extreme temperatures (from 4.63 to 23.7˚C), we were able to extract DNA from nearly all samples (DNA was not obtained from only five, < 1%, of samples) and even from specimens estimated

**Table 5. Qualitative assignment of DNA degradation in vertebrate samples used for molecular taxonomic identification.**

| Sample ID | 391 | 464 | 465 | 467 | 468 | 682 | 683 | 684 |
|---|---|---|---|---|---|---|---|---|
| DNA (ng/µl) | 11.7 | 17.7 | 230.6 | 23.8 | 10.8 | 112.0 | 15.2 | 191.0 |
| $A_{260}/A_{280}$ | 1.82 | 1.98 | 1.86 | 1.79 | 1.67 | 1.81 | 1.72 | 1.82 |
| Degradation level | 3 | 2 | 1 | 2 | 1 | 2 | 3 | 2 |
| Hours from death | 24 | 0 | 12 | 24 | 0 | 36 | 24 | 24 |

For the eight samples amplified to test molecular taxonomic identification we show the concentration (ng/µL), purity ($A_{260}/A_{280}$), and degradation level of the obtained DNA and the estimated time since death of the specimen.

**Table 6. Local alignment of *COI* sequences amplified from seven reptilian and one amphibian samples.**

| ID | Class | Query (bp) | Query cover (%) | Identity (%) | Accession N° | Reference | Match length (bp) |
|---|---|---|---|---|---|---|---|
| **391** | Reptilia | 664 | 99 | 87.18 | AB079597.1 | *Rena humilis* mitochondrial DNA | 1548 |
| **464** | Reptilia | 671 | 97 | 93.12 | MH140069.1 | *Atractus imperfectus* voucher CH:9399 *cytochrome oxidase subunit 1 (COI)* gene, partial. | 671 |
| **465** | Reptilia | 656 | 98 | 92.39 | MH140069.1 | *Atractus imperfectus* voucher CH:9399 *cytochrome oxidase subunit 1 (COI)* gene, partial. | 654 |
| **467** | Reptilia | 656 | 99 | 89.08 | MH140069.1 | *Atractus imperfectus* voucher CH:9399 *cytochrome oxidase subunit 1 (COI)* gene, partial. | 654 |
| **468** | Amphibia | 670 | 100 | 84.33 | KF540146.1 | *Caecilia tentaculata* mitochondrion | 1553 |
| **682** | Reptilia | 661 | 97 | 92.37 | MH140069.1 | *Atractus imperfectus* voucher CH:9399 *cytochrome oxidase subunit 1 (COI)* gene, partial. | 654 |
| **683** | Reptilia | 663 | 98 | 93.24 | MH140069.1 | *Atractus imperfectus* voucher CH:9399 *cytochrome oxidase subunit 1 (COI)* gene, partial. | 654 |
| **684** | Reptilia | 661 | 97 | 92.52 | MH140069.1 | *Atractus imperfectus* voucher CH:9399 *cytochrome oxidase subunit 1 (COI)* gene, partial. | 654 |

For the eight samples amplified to test molecular taxonomic identification we show the sample ID, taxonomic class, query length, percentage of the query sequence length that is included in the alignment, percentage identity reflecting the percentage of bases that are identical between the query and the reference genome (match), and details of the match sequences (detected using BLAST) in the NCBI repository.

to have been on the road for 120 hours (5 days). These results suggest this method could be implemented without intensive daily road surveys and still provide a high return of genetic material from collected tissues. While promising, the potential to obtain good quality and pure DNA may vary across regions depending on environmental conditions, mainly temperature and humidity, but also soil acidity and UV exposure [45]. Traffic levels may also be relevant as high numbers of vehicles will contribute to a faster deterioration of roadkill specimens.

We obtained DNA from specimens in all four sampled vertebrate groups, but amphibian samples returned less and lower quality DNA. While we are not sure why, it is possible that the highly permeable dermis of amphibians [46, 47] allows substances present in the environment

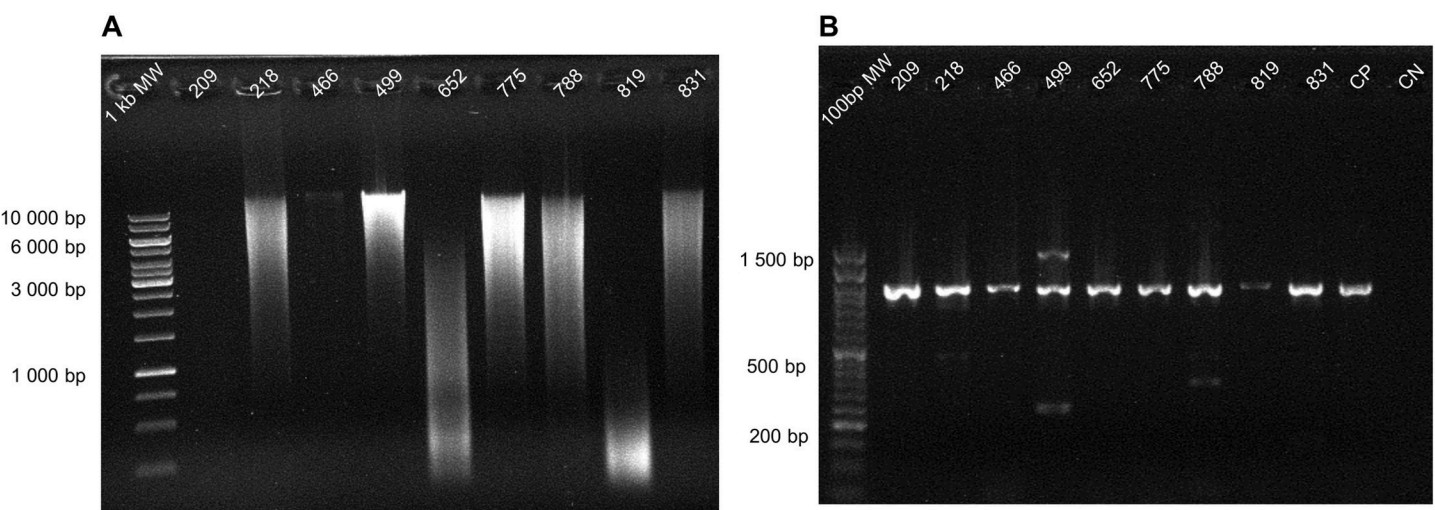

**Fig 6. Amplification of molecular target from *Leishmania* spp.** DNA electrophoresis from eight samples of vertebrate liver obtained from roadkilled specimens, containing isolated genomic DNA (A), or the amplified *18S SSU rRNA* of *Leishmania* spp. (B).

**Table 7. Results from local alignment of *18S SSU-rRNA* sequences amplified from liver samples using BLAST.**

| ID | Query (bp) | Query cover (%) | Identity (%) | Accession N° | Reference | Match length (bp) |
|---|---|---|---|---|---|---|
| **209** | 915 | 99 | 100 | GQ332354.1 | *Leishmania amazonensis* 18S ribosomal RNA gene, complete sequence; and internal transcribed spacer 1, partial sequence | 2191 |
| **218** | 908 | 99 | 99.89 | GQ332354.1 | | |
| **466** | 925 | 98 | 100 | GQ332354.1 | | |
| **499** | 919 | 99 | 100 | GQ332354.1 | | |
| **652** | 908 | 99 | 99.89 | GQ332354.1 | | |
| **775** | 911 | 99 | 100 | GQ332354.1 | | |
| **788** | 914 | 99 | 100 | GQ332354.1 | | |
| **819** | 913 | 99 | 100 | GQ332354.1 | | |
| **831** | 914 | 99 | 99.89 | GQ332354.1 | | |

For the 57 liver samples amplified to explore the molecular detection of *Leishmania* spp. we show the sample ID, query length, percentage of the query sequence length that is included in the alignment, percentage identity reflecting the percentage of bases that are identical between the query and the reference genome (match), and details of the match sequences (detected using BLAST) in the NCBI repository.

to penetrate tissues and cells degrading DNA. In addition, we also found that older (longer times since death) bird samples produced lower quantities of DNA, but this trend was not detected in other taxonomic groups. We are not sure why bird samples were more affected by time since death, but it would be interesting to explore if this pattern is also observed in other regions.

We were also able to obtain DNA from all types of tissues sampled. Liver samples produced higher DNA quantities while muscle generally yield less. Previous studies have also reported higher DNA concentrations from liver [48, 49] which may be linked to regenerating hepatocytes being tetraploids, and thus, containing more DNA [50]. Differences among tissues may be related to differences in cellular structure, cell density, and also different reactions when exposed to environmental conditions [45]. These aspects should be further study, but our initial results suggest that sampling liver, if possible, is ideal.

Not surprisingly, DNA degraded over time (since death). DNA fragmentation can interfere with the amplification of complete target genes and result in sequencing errors [51]. In our study, the observed levels of degradation did not affect amplification for vertebrate identification, but may have affected our ability to amplify parasite DNA. DNA from microorganisms is found in lower quantities within the host tissue and may be more susceptible to degradation affecting amplification [52]. To minimize this risk we used a nested PCR, a technique with high sensibility and specificity for diagnostic of visceral leishmaniasis for samples with low parasitaemia [53]. While we show this technique can be successful, we cannot rule out the possibility that parasites were present in other specimens but we could not detect them.

**Table 8. Qualitative assignment of DNA degradation from liver samples.**

| Sample ID | 209 | 218 | 466 | 499 | 652 | 775 | 788 | 819 | 831 |
|---|---|---|---|---|---|---|---|---|---|
| **DNA (ng/µl)** | 7.1 | 188.5 | 11.7 | 35.9 | 125.9 | 123.8 | 55.3 | 38.1 | 261.2 |
| **$A_{260}/A_{280}$** | 1.52 | 1.85 | 1.87 | 1.66 | 1.71 | 1.80 | 1.76 | 1.68 | 1.77 |
| **Degradation level** | 3 | 2 | 1 | 1 | 3 | 2 | 2 | 3 | 2 |
| **Hours from death** | 0 | 24 | 6 | 0 | 6 | 12 | 12 | 1 | 24 |

For the 57 liver samples amplified to explore the molecular detection of *Leishmania* spp. we show the concentration (ng/µL), purity ($A_{260}/A_{280}$), and degradation level of the obtained DNA and the estimated time since death of the specimen.

Overall, our study shows that DNA extracted from carcasses of roadkill animals can be of sufficient quality for downstream applications. Our findings are consistent with conclusions from studies considering postmortem DNA degradation in forensic samples, for example to identify missing people or sequence museum samples [5, 25]. Biobanks of tissues and DNA can be generated from samples collected from roadkill animals without adding an impact on wildlife. These data can help us answer questions about species distributions, diseases, taxonomy, and generally contribute to preserve biological information. Although it is crucial to reduce the negative impacts of roads on wildlife [54], roads are necessary and roadkill will never be completely avoidable. Making the most of a bad situation, roadkill carcasses should be collected and treated as precious samples that can help us gain a better understanding of local biodiversity, especially in countries like Ecuador that have high diversity and rates of endemism, but where much is yet to be learned.

## Supporting information

**S1 Fig. Electrophoresis gel showing genomic DNA isolated from intestine samples.** The sample ID is shown at the top. The bottom labels indicate our qualitative classification: "1" (minor degradation), "2" (medium degradation) o "3" (high degradation).
(TIF)

**S1 Raw images. Original images of the electrophoresis gels shown in Figs 5B, 6B, and S1.**
(PDF)

**S1 Table. Alternative regression models testing how DNA quantity and purity varied by class and time since death.**
(DOCX)

**S2 Table. Qualitative assignment of DNA degradation.** Samples were classified as "1" (minor degradation), "2" (medium degradation) o "3" (high degradation).
(DOCX)

**S1 File. Database of all samples analyzed in this study.** Details of the samples collected during the roadkill survey in the Napo region of Ecuador from 2020 to 2021.
(XLSX)

**S2 File. Sequences obtained in this study.** Sequences of *COI* and *18S* from *Leishmania* spp. obtained from the amplification of specific targets by PCR.
(FASTA)

## Acknowledgments

We thank F. Vaca, R. Rodríguez and V. Herrera from Universidad Central del Ecuador and X. Fiallos from Yachay, who helped us in implementing this project.

## Author Contributions

**Conceptualization:** Sandra Enríquez, Sarah Martin-Solano, Sofía Ocaña-Mayorga, Gabriel Alberto Carrillo-Bilbao, Manuela González-Suárez, Ana Poveda.

**Formal analysis:** Manuel Alejandro Coba-Males, Sarah Martin-Solano, Sofía Ocaña-Mayorga, Jazzmín Arrivillaga-Henríquez, Manuela González-Suárez, Ana Poveda.

**Funding acquisition:** Sandra Enríquez, Sofía Ocaña-Mayorga, Wilmer Narváez, Manuela González-Suárez, Ana Poveda.

**Investigation:** Manuel Alejandro Coba-Males, Pablo Medrano-Vizcaíno, David Brito-Zapata, Jaime Antonio Salas.

**Methodology:** Manuel Alejandro Coba-Males, Pablo Medrano-Vizcaíno, David Brito-Zapata, Sarah Martin-Solano, Sofía Ocaña-Mayorga, Gabriel Alberto Carrillo-Bilbao, Manuela González-Suárez, Ana Poveda.

**Project administration:** Sandra Enríquez, Wilmer Narváez, Manuela González-Suárez, Ana Poveda.

**Supervision:** Sandra Enríquez, Sarah Martin-Solano, Sofía Ocaña-Mayorga, Gabriel Alberto Carrillo-Bilbao, Manuela González-Suárez, Ana Poveda.

**Writing – original draft:** Manuel Alejandro Coba-Males, Pablo Medrano-Vizcaíno, Sandra Enríquez, David Brito-Zapata, Sarah Martin-Solano, Sofía Ocaña-Mayorga, Gabriel Alberto Carrillo-Bilbao, Jaime Antonio Salas, Jazzmín Arrivillaga-Henríquez, Manuela González-Suárez, Ana Poveda.

**Writing – review & editing:** Manuel Alejandro Coba-Males, Pablo Medrano-Vizcaíno, Sandra Enríquez, David Brito-Zapata, Sarah Martin-Solano, Sofía Ocaña-Mayorga, Gabriel Alberto Carrillo-Bilbao, Jaime Antonio Salas, Jazzmín Arrivillaga-Henríquez, Manuela González-Suárez, Ana Poveda.

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
