## [Decision Letter · Decision Letter 0]

2 May 2023

PONE-D-23-09739Roadkill animals as valuable source of genetic data to monitor the diversity of vertebrates and their associated microorganism without additional impact on wildlifePLOS ONE

Dear Dr. Gonzalez-Suarez,

Thank you for submitting your manuscript to PLOS ONE. After careful consideration, we feel that it has merit but does not fully meet PLOS ONE’s publication criteria as it currently stands. Therefore, we invite you to submit a revised version of the manuscript that addresses the points raised during the review process. Please submit your revised manuscript by Jun 16 2023 11:59PM. If you will need more time than this to complete your revisions, please reply to this message or contact the journal office at plosone@plos.org. Please include the following items when submitting your revised manuscript:A rebuttal letter that responds to each point raised by the academic editor and reviewer(s). You should upload this letter as a separate file labeled 'Response to Reviewers'.A marked-up copy of your manuscript that highlights changes made to the original version. You should upload this as a separate file labeled 'Revised Manuscript with Track Changes'.An unmarked version of your revised paper without tracked changes. You should upload this as a separate file labeled 'Manuscript'.If applicable, we recommend that you deposit your laboratory protocols in protocols.io to enhance the reproducibility of your results. Protocols.io assigns your protocol its own identifier (DOI) so that it can be cited independently in the future. For instructions see: https://journals.plos.org/plosone/s/submission-guidelines#loc-laboratory-protocols. Additionally, PLOS ONE offers an option for publishing peer-reviewed Lab Protocol articles, which describe protocols hosted on protocols.io. Read more information on sharing protocols at https://plos.org/protocols?utm_medium=editorial-email&utm_source=authorletters&utm_campaign=protocols.

We look forward to receiving your revised manuscript.

Kind regards,

Ruslan Kalendar

Academic Editor

PLOS ONE

Journal Requirements:

"The authors would like to thank to: Corporación Ecuatoriana para el Desarrollo de la Investigación y Academia - CEDIA for the financial support given to the present research, development, and innovation work through its CEPRA program, especially for the Estudio de parásitos y microbioma de fauna silvestre en dos de las zonas más biodiversas del planeta: Los Andes Tropicales y Chocó-Darién en Ecuador fund (https://cedia.edu.ec/servicio/fondo-idi-universidades/cepra-xvi-2022/); Dirección de Investigación de la Universidad Central del Ecuador for Proyecto Senior 2021 fund (https://www.uce.edu.ec/web/di), and Reading University for SBS Seed Fund and a University International PhD studentship to PMV (" ext-link-type="uri" xlink:type="simple">https://www.reading.ac.uk/"

7. We note that Figure 1 in your submission contain map/satellite image which may be copyrighted. All PLOS content is published under the Creative Commons Attribution License (CC BY 4.0), which means that the manuscript, images, and Supporting Information files will be freely available online, and any third party is permitted to access, download, copy, distribute, and use these materials in any way, even commercially, with proper attribution. For these reasons, we cannot publish previously copyrighted maps or satellite images created using proprietary data, such as Google software (Google Maps, Street View, and Earth). For more information, see our copyright guidelines: http://journals.plos.org/plosone/s/licenses-and-copyright.

Reviewers' comments:

Reviewer's Responses to Questions

**Comments to the Author**

1. Is the manuscript technically sound, and do the data support the conclusions?

Reviewer #1: Yes

Reviewer #2: Yes

2. Has the statistical analysis been performed appropriately and rigorously? 

Reviewer #1: Yes

Reviewer #2: Yes

3. Have the authors made all data underlying the findings in their manuscript fully available?

Reviewer #1: Yes

Reviewer #2: Yes

4. Is the manuscript presented in an intelligible fashion and written in standard English?

Reviewer #1: Yes

Reviewer #2: Yes

5. Review Comments to the Author

**Reviewer #1**: 

Please see comments in the PDF file.

In general, the main purpose of this study is interesting and meaningful (especially when many tropical countries with high biodiversity but low budgets and low number of ecological studies), data collection method and analysis approaches are clear and good, and discussion and conclusion are relevant to the findings.

**Reviewer #2**: 

The work is interesting, but the content examines the effects of using animals killed on the road for DNA isolation. The influence of the time since death and the type of tissue on the quality of the obtained DNA is being studied. However, there is no assessment of diversity, both between species and within species. So the work is very good, but it is necessary to change the title to be adequate to the content. There is no results or discussion about “monitor the diversity of vertebrates” and also only one microorganism was check in the DNA. May be something like "quality of DNA isolated from road killed animals depeding on time after death and kind of tissue" - or other because in this title there is nothing about microorganism.

The small comment about Line 112 – are values of precipitation and temperatures annual? –an average for a year? Or this values are for shorter period within year? The question is because of large differences

6. PLOS authors have the option to publish the peer review history of their article (what does this mean?). If published, this will include your full peer review and any attached files.

Reviewer #1: No

Reviewer #2: No

---

## [Author Response · Author response to Decision Letter 0]

14 Aug 2023

A response letter has been uploaded

---

## [Editor Report · Decision Letter 1]

16 Aug 2023

From roads to biobanks: roadkill animals as a valuable source of genetic data

PONE-D-23-09739R1

Dear Dr. Gonzalez-Suarez,

We’re pleased to inform you that your manuscript has been judged scientifically suitable for publication and will be formally accepted for publication once it meets all outstanding technical requirements.

Kind regards,

Ruslan Kalendar

Academic Editor

PLOS ONE

---

## [Editor Report · Acceptance letter]

29 Aug 2023

PONE-D-23-09739R1 

From roads to biobanks: roadkill animals as a valuable source of genetic data 

Dear Dr. González-Suárez:

I'm pleased to inform you that your manuscript has been deemed suitable for publication in PLOS ONE. Congratulations! Your manuscript is now with our production department. 

Kind regards, 

on behalf of

Professor Ruslan Kalendar 

Academic Editor

PLOS ONE